# Comparative Study of the Priming Effect of Abscisic Acid on Tolerance to Saline and Alkaline Stresses in Rice Seedlings

Zhonghui Feng [1,2,3,†] (ID), Guanru Lu [1,3,†], Miao Sun [2], Yangyang Jin [1], Yang Xu [1,4], Xiaolong Liu [5], Mingming Wang [1,4], Miao Liu [1,4], Haoyu Yang [1,4], Yi Guan [2], Tianhe Yu [1,4], Jiafeng Hu [1,4], Zhiming Xie [2], Weiqiang Li [1,4,*] (ID) and Zhengwei Liang [1,4,*]

1    State Key Laboratory of Black Soils Conservation and Utilization, Northeast Institute of Geography and Agroecology, Chinese Academy of Sciences, Changchun 130102, China; fengzhonghui@iga.ac.cn (Z.F.); luguanru@iga.ac.cn (G.L.); xuyang@iga.ac.cn (Y.X.); wangmingming@iga.ac.cn (M.W.); liumiao@iga.ac.cn (M.L.); yanghaoyu@iga.ac.cn (H.Y.)
2    College of Life Science, Baicheng Normal University, Baicheng 137000, China; xiezhiming@bcnu.edu.cn (Z.X.)
3    University of Chinese Academy of Sciences, Beijing 100049, China
4    Jilin Da'an Agro-Ecosystem National Observation Research Station, Changchun Jingyuetan Remote Sensing Experiment Station, Da'an 131317, China
5    College of Life Sciences, Resources and Environment, Yichun University, Yichun 336000, China; lxl190092@jxycu.edu.cn
*    Correspondence: liweiqiang@iga.ac.cn (W.L.); liangzw@iga.ac.cn (Z.L.); Tel.: +86-431-8554-2347 (Z.L.)
†    These authors contributed equally to this work.

**Abstract:** The plant hormone abscisic acid (ABA) regulates the plant response to environmental stress; therefore, ABA priming is an effective strategy for enhancing stress tolerance in rice. In this study, we investigated the priming effects of 1 and 5 μM ABA on the biochemical and physiological traits associated with seedling growth performance in two rice cultivars exposed to saline (100 mM NaCl) and alkaline (15 mM $Na_2CO_3$) stress via root drenching. ABA pretreatment effectively reduced damage in rice seedlings by mitigating the increases in $Na^+/K^+$ ratio, membrane injury, contents of $Na^+$, malondialdehyde, hydrogen peroxide, and superoxide anion radical, and prevented reductions in $K^+$ and total chlorophyll contents, and ROS-related enzyme activities in both cultivars under saline and alkaline stresses. Rice seedlings with ABA pretreatment under alkaline stress had a stronger ability to maintain ion homeostasis, eliminate ROS, and induce changes in endogenous ABA levels via the upregulation of *OsHKT1;5*, *OsSOS1*, *OsNHX5*, *OsPOX1*, *OsCATA*, *OsNCED3*, *OsSalT*, and *OsWsi18* and downregulation of *OsRbohA* than under saline stress. The saline–alkaline (SA)-sensitive cultivar demonstrated greater sensitivity to the priming effect of ABA than that of the SA-tolerant cultivar under both stress conditions. These findings have implications for rice adaptation to SA soils.

**Keywords:** abscisic acid; priming; rice; tolerance to saline and alkaline stresses





## 1. Introduction

The salinization and alkalinization of soil, leading to saline–alkaline (SA) stress, are significant constraints on crop productivity, affecting approximately 20% of irrigated agricultural land worldwide [1–3]. The Food and Agriculture Organization of the United Nations [4] reported that the global land area affected by salinization and alkalinization has surpassed 830 million hectares, which accounts for approximately 60% of the estimated sodic area characterized by high pH levels due to elevated concentrations of sodium bicarbonate ($NaHCO_3$) and sodium carbonate ($Na_2CO_3$). Secondary salinity affects around 25 to 33% of irrigated land worldwide. Furthermore, the annual growth rate is greater than $1 \times 10^6$ hectares [5]. The high salinity and pH levels associated with SA stress pose a greater threat to plants than that of either stress alone [6,7]. SA stress can inhibit plant growth via osmotic stress [8–10] and ion toxicity (mainly $Na^+$ and $Cl^-$) [11–13]. In contrast to the impact of saline stress (pH~7) alone, alkaline stress (high pH ranging from 8.5 to 11.0) has

the potential to disrupt physiological metabolism [14–16], leading to root damage [17–19] and a nutritional imbalance [20–22]. These effects can have detrimental consequences on plant growth by inducing heightened cellular oxidative stress [23]. The cells and cellular compartments of plants possess a distinctive system that ensures the precise parameters for growth and differentiation, thereby facilitating normal tissue development [24]. The restriction of growth caused by salinity and alkalinity is associated with the disruption of plastids and mitochondria functioning, along with proliferation and elongation growth [25,26].

The cultivation of rice (*Oryza sativa* L.) under adequately irrigated conditions is a viable approach for enhancing grain production in SA soil [27]. However, rice is highly susceptible to SA stress [28]. SA stress significantly inhibits seedling survival and growth [29], causing severe damage to cells, and even leading to the death of the entire plant [30,31], resulting in poor grain yield and quality [32,33].

Priming (i.e., the induction of increased plant tolerance to future stressors) can be achieved via exposure to biotic or abiotic stresses as well as the application of plant hormones and their synthetic analogues with functional properties [34–36]. The plant hormone abscisic acid (ABA) plays a pivotal regulatory role in modulating plant responses to environmental stress [37–39]. This hormone has a remarkable priming effect on plants by activating various defense mechanisms and signaling pathways, while modifying gene expression to assist plants in adapting to potential challenges [40,41]. The acclimation of plants to stress is regulated by the ABA signaling pathway, which contributes to the perception and transmission of signals [42,43]. A comprehensive understanding of the mechanisms underlying ABA priming in the regulation of stress tolerance in crops would greatly facilitate molecular plant breeding [44]. Although numerous studies have evaluated ABA resistance to various abiotic stresses, few comparative studies have specifically examined the effect of ABA priming on the tolerance to different stress conditions. Therefore, the elucidation of the priming effect of ABA on the response to saline and alkaline stresses could be helpful for resistant rice cultivation.

In this study, we conducted a range of physiological, biochemical, and molecular investigations to analyze the characteristics and reactions of two cultivars of rice under saline and alkaline stresses to (i) describe the different responses of the cultivars to saline and alkaline stresses and (ii) compare the contribution of ABA to the saline and alkaline tolerance of rice seedlings based on its effects on physiological and biochemical processes and gene expression. This study provides a comparative analysis of the mechanisms underlying the effects of ABA priming on the response to saline and alkaline stress conditions.

## 2. Materials and Methods

### 2.1. Plant Material and Growth Conditions

Two local rice cultivars were used in the experiment: "Dongdao-4" (D-4, tolerant to SA stress) and "Jiudao-51" (J-51, sensitive to SA stress) [19]. The rice seeds under examination were subjected to disinfection using a 0.1% (*w/v*) $HgCl_2$ solution for 10 min, followed by thorough rinsing with distilled water. The seeds were spread on the bottom of a plastic Petri dish and germinated in an incubator at 30 °C for 24 h. Rice seeds with the same germination were selected and seeded as a group of 20 individuals into a 4 cm × 5 cm plastic floating net. The floating net was placed into a 330 mL plastic cup for 7 d, and the seeds were grown for another 7 d with half-strength Kimura B nutrient solution [45] in an artificial climate chamber (HPG-400HX; Hadonglian Inc., Beijing, China) at 25 °C (light)/20 °C (dark) and a 12 h photoperiod.

### 2.2. Stress Treatment and ABA Application

Two-week-old rice seedlings were root-drenched with 0, 1, and 5 μM ABA (obtained from Sigma, Inc., St. Louis, MO, USA) for 24 h and then incubated in 100 mM NaCl (pH: 6.97 ± 0.021, EC: 10.09 ± 0.002 mS/cm) and 15 mM $Na_2CO_3$ (pH: 10.82 ± 0.032, EC: 2.80 ± 0.002 mS/cm) to simulate saline and alkaline stress conditions, respectively. S0, S1, and S5 and A0, A1, and A5 represented rice seedlings pretreated with 0, 1, and



5 µM ABA under saline and alkaline stresses, respectively. CK was the unstressed control (nutrient solution without ABA). A pH meter (pH-25; Lida Inc., Shanghai, China) and a conductivity meter (DDS-12; Baiyuan Inc., Beijing, China) were used to measure the pH and electrical conductivity (EC) of the solutions, respectively.

### 2.3. Measurement of Seedling Growth

After 6 d of saline and alkaline stress, the survival rate of each treated rice seedling was investigated. If the leaves were shriveled and brown, the seedlings were considered dead. Ten seedlings were randomly selected to measure the plant height. The seedlings were deactivated at 105 °C for 1 h, followed by 65 °C until reaching a steady mass to measure the shoot and root dry weights.

### 2.4. Measurement of $Na^+$ and $K^+$ Contents

After determining the dry weight, the sample was cut into fragments, subjected to digestion with a mixture of $HNO_3$:$HClO_4$ ($v/v$ = 2:1), and diluted to a final volume of 50 mL. The shoot and root $Na^+$ and $K^+$ contents were determined using a flame photometer (FP6410; Shanghai Precision and Scientific Instrument Co., Ltd., Shanghai, China).

### 2.5. Measurement of the Chlorophyll Content

A 10 mL mixture of ethanol and acetone ($v/v$ = 1:1) was used to extract leaf samples from rice seedlings. The absorbance of the supernatant was measured at 663 and 645 nm. The chlorophyll content was determined using the following formula: Total chlorophyll content = (8.05 $A_{663}$ + 20.29 $A_{645}$) V/1000 W.

### 2.6. Measurement of Membrane Injury (MI) and Malondialdehyde (MDA) Contents

MI was evaluated quantitatively by measuring the relative electrolyte leakage [19] and calculated using the formula MI (%) = R1/R2 × 100. The conductivity of tissue samples was measured before (R1) and after boiling (R2) in water. The MDA content was quantified using the thiobarbituric acid reaction method [46] and determined using the formula $6.45 \times 242$ ($A_{532} - A_{600}$) $- 0.56 \times A_{450}$.

### 2.7. Measurement of Superoxide Anion Radical ($O_2 \cdot^-$) and Hydrogen Peroxide ($H_2O_2$) Levels

The $O_2 \cdot^-$ content was measured by monitoring the formation of nitrite from hydroxylamine in the presence of $O_2 \cdot^-$, as described by Jiang and Zhang [47]. Absorbance at 530 nm was calibrated to calculate the $O_2 \cdot^-$ content in the chemical reaction of $O_2 \cdot^-$ and hydroxylamine.

The $H_2O_2$ content was measured by monitoring the $A_{415}$ of the titanium peroxide complex [48]. The analytical reagents for measuring $H_2O_2$ and $O_2 \cdot^-$ contents were obtained using an assay kit (Comin Biotech Co., Ltd., Suzhou, China), according to the manufacturer's instructions [19].

### 2.8. Measurement of Antioxidant Enzyme Activities

The fresh leaf samples (0.1 g) were aseptically placed into 2 mL tubes, rapidly frozen in liquid nitrogen, and subsequently homogenized in 1 mL of extraction buffer following previously established protocols. The clear supernatants were used to measure activities of SOD (EC 1.15.1.1) [49], CAT (EC 1.11.1.6.) [50], POD (EC 1.11.1.7) [51], and APX (EC 1.11.1.11) [52].

### 2.9. Quantitative Real-Time Polymerase Chain Reaction (qRT-PCR)

qRT-PCR was used to determine the transcriptional expression of genes involved in ABA biosynthesis, ABA response, ion homeostasis, ROS metabolism, and stress tolerance. The leaf samples were randomly collected from each treatment after 24 h of exposure to saline and alkaline stress. Subsequently, they were pulverized in liquid nitrogen using a benchtop ball mill operating at a frequency of 50 Hz for a duration of 30 s. The total

RNA was extracted using TRIzol reagent (TaKaRa Bio, Tokyo, Japan), reverse transcribed with M-MLV reverse transcriptase (Thermo Fisher Scientific, Carlsbad, CA, USA), and amplified by PCR reactions using gene-specific primers designed with Primer 5.0 software. The housekeeping gene *β-ACTIN* (GenBank ID: X15865.1) was utilized as the reference gene. PCR reactions were conducted using a mixture comprising cDNA template, specific forward and reverse primers, SYBRR Premix Ex Taq (TaKaRa Bio), and double-distilled $H_2O$ on a PCR max machine (Eco TM48, Illumina, Saffron Walden, UK). The cycling program consisted of initial denaturation at 95 °C for 30 s followed by 30 cycles at 95 °C for 5 s and 58 °C for 34 s with a final melting curve stage at 95 °C for 15 s and 60 °C for 1 min, and 95 °C for 15 s. The relative expression level was calculated using the $2^{-\Delta\Delta CT}$ method [53], with three biological replicates per treatment. All primers used for qRT-PCR analyses are listed in Table S1.

### 2.10. Statistical Analyses

The statistical software SPSS (version 21.0; IBM Corp., Armonk, NY, USA) was utilized for conducting the statistical analyses. Duncan's multiple range test was performed based on the results obtained from one-way analysis of variance ($p < 0.05$).

## 3. Results

### 3.1. ABA Priming Increased Rice Seedling Survival under Both Saline and Alkaline Conditions

Saline and alkaline stresses over a period of 6 d resulted in the withering and mortality of seed lings from both cultivars, with J-51 exhibiting a more pronounced reduction in growth than that of D-4 (Figure 1A). Rice seedlings treated with ABA exhibited significantly higher survival rates than those of seedlings not treated with ABA (Figure 1B). Under saline stress, the survival rates of J-51 and D-4 under S1 treatment exhibited increases of 111.11% and 110.71%, respectively, compared with those under S0 treatment. Similarly, under S5 treatment, the survival rate showed a significant improvement, with increases of 88.89% for J-51 and 85.71% for D-4 compared to that under S0 treatment. Under alkaline stress, the survival rates of J-51 and D-4 under A1 treatment exhibited significant increases of 181.82% and 53.57%, respectively, over those under A0. Similarly, the survival rates of J-51 and D-4 under A5 treatment demonstrated remarkable increases of 227.27% and 64.29%, respectively, compared to those under A0.

### 3.2. ABA Priming Enhanced Rice Seedling Growth under Saline and Alkaline Stresses

We measured plant height, biomass, and chlorophyll contents to investigate the effects of ABA pretreatment on the growth and physiological characteristics of rice seedlings under saline and alkaline stresses. The plant heights of J-51 and D-4 were lower under saline stress than in the CK group, whereas the difference between groups was not significant under alkaline stress (Figure 2A). Pretreatment with 1 μM ABA resulted in a significantly higher plant height in both J-51 and D-4 than in S0 treatment under saline stress. The two rice cultivars exhibited a significant decrease in the total chlorophyll content under both saline and alkaline stresses, with a higher degree of inhibition observed under alkaline stress than saline stress. Exogenous ABA pretreatment significantly increased the chlorophyll contents of J-51 and D-4 plants under both stress conditions (Figure 2B). The chlorophyll content of D-4 was higher than that of J-51 under the different treatments.

Furthermore, the dry weights of shoots and roots of J-51 plants were lower than those of CK plants under both stress conditions. Pretreatment with 1 μM ABA significantly increased the dry weight of shoots and roots of J-51 compared to S0 treatment, while there was no appreciable effect in D-4 plants. The dry weights of the shoots of J-51 and D-4 showed no significant differences under alkaline stress. Compared to those under A0 treatment, ABA pretreatment significantly increased the dry weights of roots of J-51 and D-4 ($p < 0.05$) (Figure 2C,D).

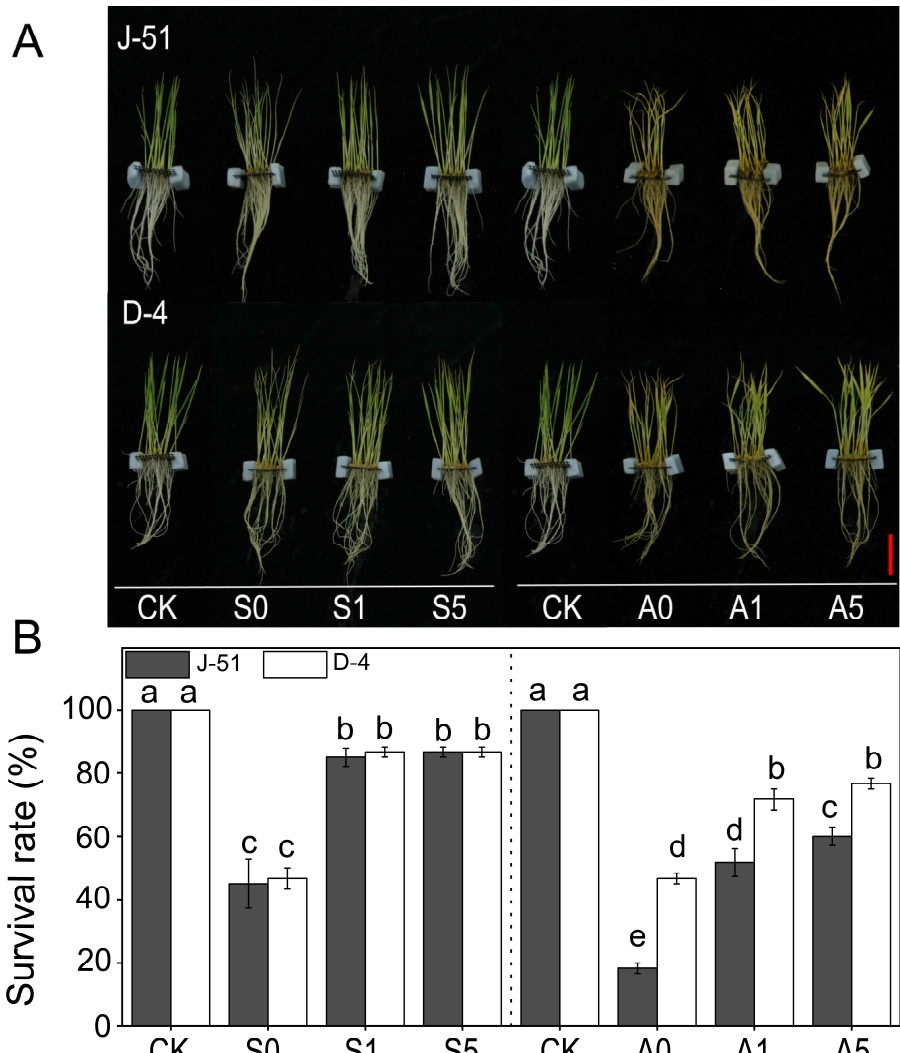

**Figure 1.** Abscisic acid (ABA) priming enhances rice seedlings growth under saline and alkaline stresses. Pictures (**A**) and survival rates (**B**) of 14 d old rice seedlings of Jiudao 51 (J-51) and Dongdao 4 (D-4) cultivars grown under unstressed control (CK, nutrient solution with non-ABA priming), 100 mM NaCl (S0, S1, S5) and 15 mM $Na_2CO_3$ (A0, A1, A5) for 6 days with the root-drenched priming of 0, 1 and 5 µM ABA for 24 h, respectively. Bar = 5 cm. Values are means ± SD, *n* = 3. Different letters on the column represent significant differences based on Ducan's test (*p* < 0.05).

### 3.3. ABA Priming Mediated Ion Homeostasis in Rice Seedlings under Saline and Alkaline Stresses

Exposure to saline and alkaline stresses resulted in significant increases in the $Na^+$ content and the $Na^+/K^+$ ratio, accompanied by a notable reduction in the $K^+$ concentration in both the shoots and roots of J-51 and D-4. Pretreatment with 1 µM ABA led to a significant reduction in the $Na^+$ content and $Na^+/K^+$ ratio as well as a slight increase in the $K^+$ content in both shoots and roots of J-51 and D-4 plants subjected to saline and alkaline stresses, respectively (Figure 3A,B) (*p* < 0.05), and the effect of 5 µM ABA was greater than that of 1 µM ABA. Pretreatment with 5 µM ABA significantly reduced the shoot $Na^+/K^+$ ratio by 47.20% in J-51 and 30.47% in D-4 under saline stress as well as by 5.02% in J-51 and 9.33% in D-4 under alkaline stress; it also reduced the root $Na^+/K^+$ ratio by 25.76% in J-51 and 15.04% in D-4 under saline stress as well as by 33.52% in J-51 and 21.82% in D-4 under alkaline stress (*p* < 0.05).

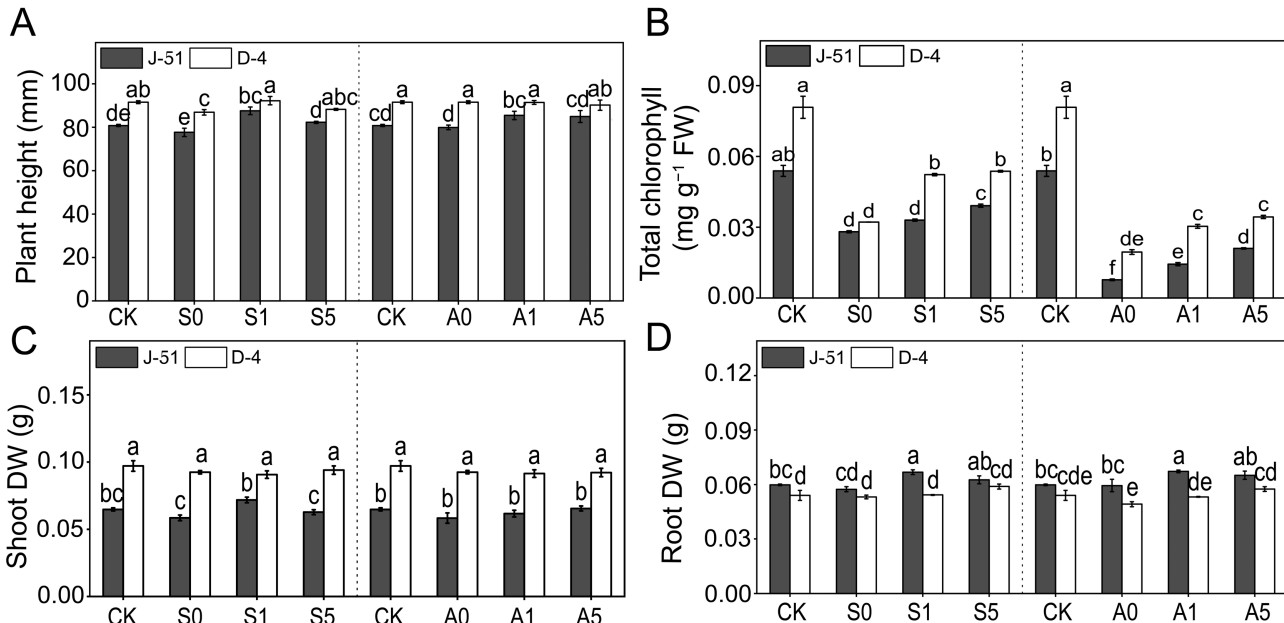

**Figure 2.** Effect of ABA priming on dry weight (DW), plant height and chlorophyll content in rice seedlings under saline and alkaline stresses conditions. Plant height (A), Total chlorophyll (B), Shoot DW (C) and Root DW (D) of 14 d old rice seedlings of Jiudao 51 (J-51) and Dongdao 4 (D-4) grown for 6 days under unstressed control (CK, nutrient solution with non-ABA), 100 mM NaCl (S0, S1, S5) and 15 mM $Na_2CO_3$ (A0, A1, A5) with the root-drenched priming of 0, 1 and 5 µM ABA for 24 h, respectively. Values are means ± SD, $n$ = 3. Different letters on the column represent significant differences based on Ducan's test ($p$ < 0.05).

### 3.4. ABA Priming Mitigated Plasma Membrane Damage and Reduced MDA and ROS Accumulation under Saline and Alkaline Stresses

MI in shoots and roots was greater under alkaline stress than under saline stress. Pretreatment with 5 µM ABA significantly reduced shoot MI by 20.37% in J-51 and 28.36% in D-4 under saline stress as well as by 27.58% in J-51 and 30.59% in D-4 under alkaline stress; it reduced root MI by 15.04% in J-51 and 19.26% in D-4 under saline stress as well as by 25.13% in J-51 and 21.09% in D-4 under alkaline stress ($p$ < 0.05) (Figure 4A). Significant accumulation of MDA, $H_2O_2$, and $O_2 \cdot^-$ was observed in both cultivars in response to saline and alkaline stresses. The levels of MDA, $H_2O_2$, and $O_2 \cdot^-$ under alkaline stress were considerably higher than those under saline stress, with J-51 exhibiting higher levels of MDA, $H_2O_2$, and $O_2 \cdot^-$ than those in D-4. ABA pretreatment resulted in significantly reduced MDA, $H_2O_2$, and $O_2 \cdot^-$ contents in J-51 and D-4 under both stresses ($p$ < 0.05) (Figure 4B).

### 3.5. ABA Priming Alleviated Oxidative Damage by Modulating the Activities of Antioxidant Enzymes under Saline and Alkaline Stresses

Antioxidant defense mechanisms are activated by rice seedlings in response to the stress-induced accumulation of reactive oxygen species (ROS) to eliminate excessive ROS from cells. Compared to levels in CK, saline and alkaline stresses triggered increases in the SOD, POD, CAT, and APX activities in J-51 and D-4 plants. Similarly, exogenous ABA pretreatment led to significant increases in SOD, POD, CAT, and APX activities in J-51 and D-4 plants under saline and alkaline stresses. Notably, the effect of 5 µM ABA pretreatment on J-51 under alkaline stress was slightly more pronounced than that of 1 µM ABA pretreatment under both stress conditions (Figure 5) ($p$ < 0.05).

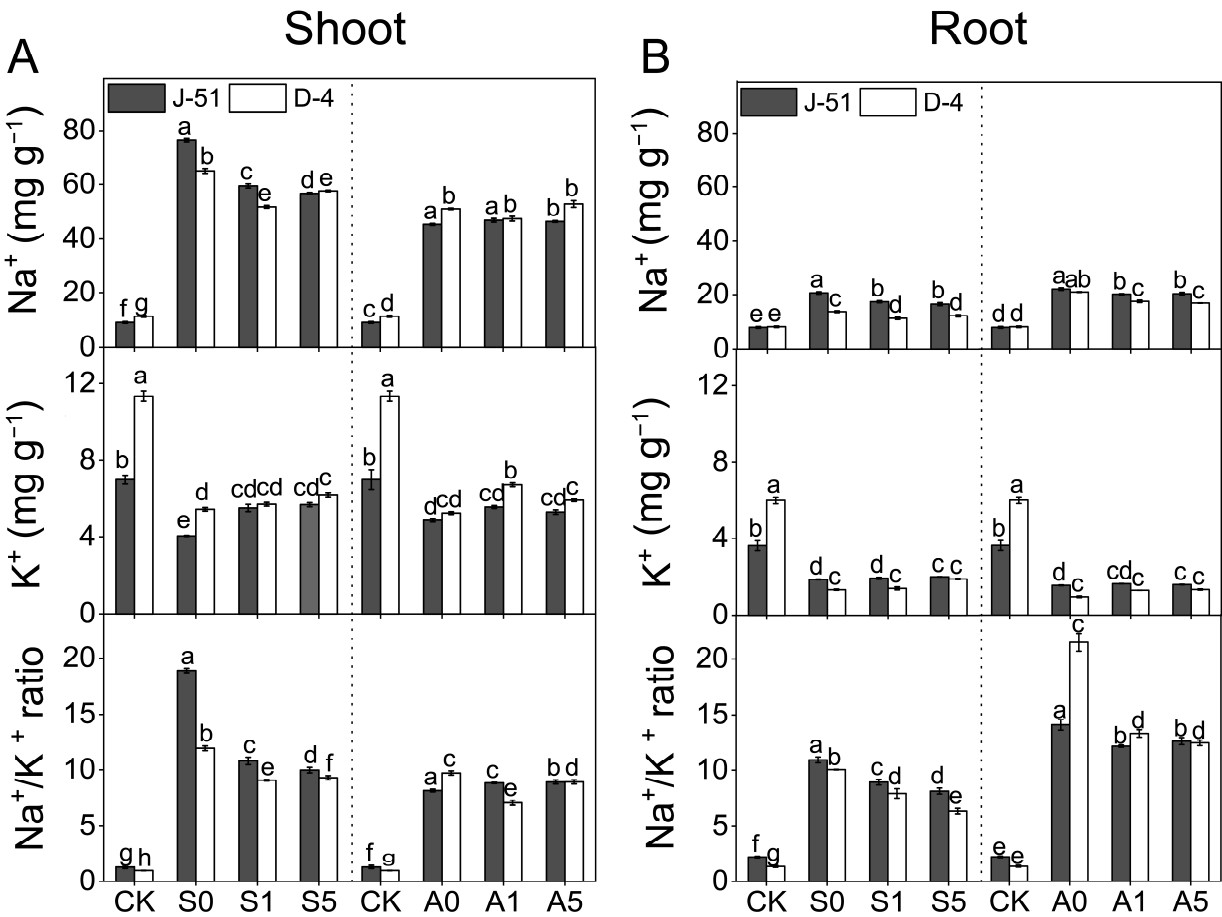

**Figure 3.** Effects of ABA priming on levels of $Na^+$ and $K^+$, and $Na^+/K^+$ ratio in rice seedlings under saline and alkaline stresses conditions. Contents of $Na^+$ and $K^+$, and $Na^+/K^+$ ratio of shoots (**A**) and roots (**B**) in 14 d old rice seedlings of Jiudao 51 (J-51) and Dongdao 4 (D-4) grown for 6 days under unstressed control (CK, nutrient solution with non-ABA), 100 mM NaCl (S0, S1, S5) and 15 mM $Na_2CO_3$ (A0, A1, A5) with the root-drenched priming of 0, 1 and 5 μM ABA for 24 h, respectively. Values are means ± SD, *n* = 3. Different letters on the column represent significant differences based on Ducan's test (*p* < 0.05).

### 3.6. Multivariate Statistical Analysis of the Rice Seedling Response to ABA Application under Saline and Alkaline Stresses

A principal component analysis (PCA) was performed based on all relevant data to explain the patterns of rice seedling responses to ABA application under saline and alkaline stresses. The growth, physiological, biochemical, and mineral traits of each cultivar are shown in biplots. Under both saline and alkaline stresses, the treatments were well separated in the score plots (Figure 6A–D). Under saline stress, PC1 and PC2 in J-51 accounted for 65.4% and 22.5% of the total variance, respectively, while PC1 and PC2 in D-4 accounted for 69.0% and 13.8%, respectively. PC1 reflected variance between non-saline and saline conditions, whereas PC2 displayed variance between ABA-treated and non-ABA-treated rice seedlings. The growth-related physiological parameters (root DW and root MI), mineral traits (root $Na^+$ content, shoot $Na^+$ content and root $Na^+/K^+$ ratio), and enzymatic antioxidants (CAT, APX, SOD, and POD) were mostly associated with ABA-treated J-51 rice seedlings exposed to saline stress, whereas the growth-related mineral traits (root $Na^+$ content) and enzymatic antioxidants (CAT, APX, SOD, and POD) were linked with ABA-treated D-4 rice seedlings under saline stress conditions (Figure 6A,B).

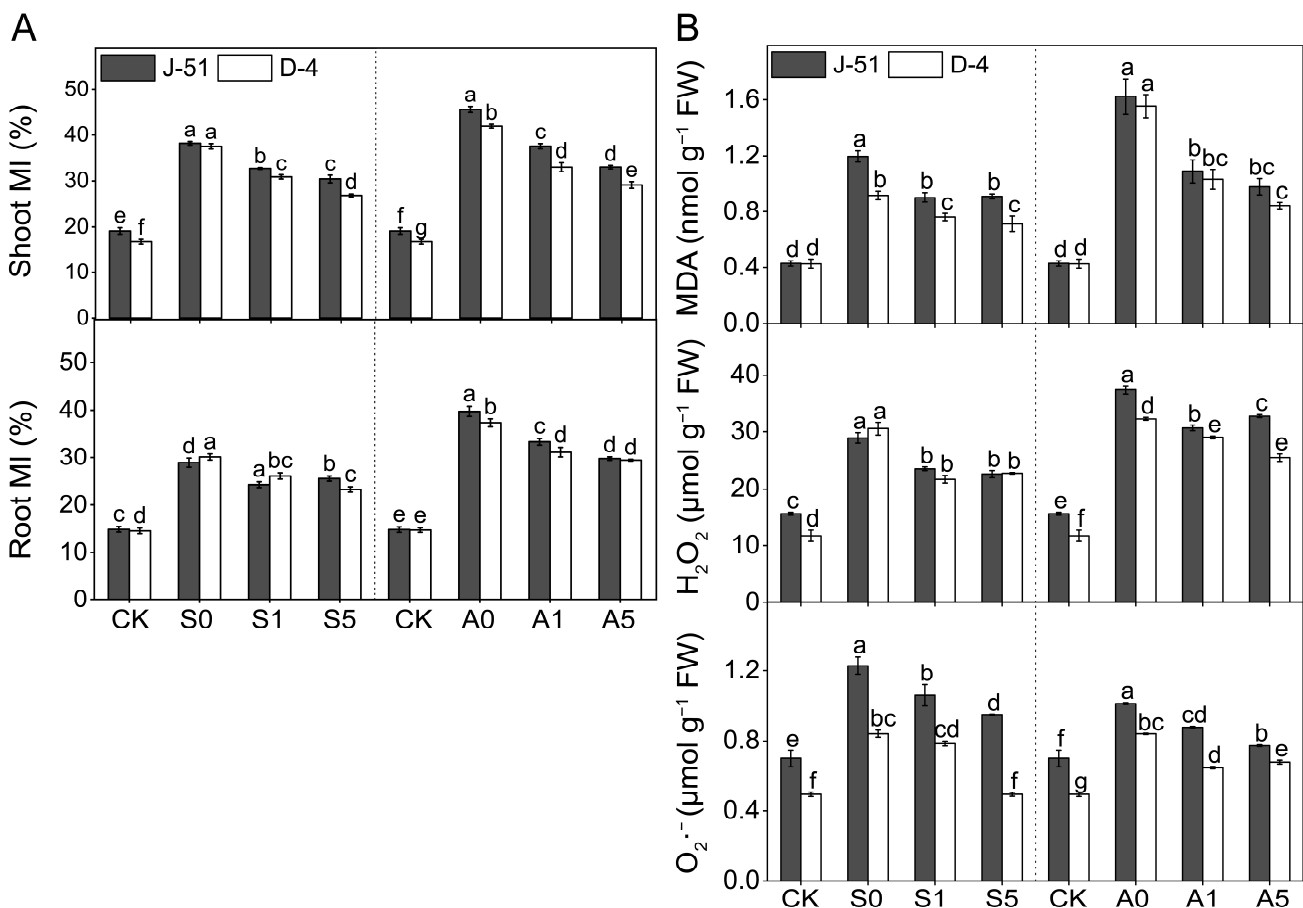

**Figure 4.** Effects of ABA priming on levels of membrane injury (MI) and state of reactive oxygen species in rice seedlings under saline and alkaline stress conditions. MI of shoots and roots (**A**) and concentrations of malondialdehyde (MDA), hydrogen peroxide ($H_2O_2$), and super oxide anion radicals ($O_2 \cdot^-$) (**B**) in leaves of 14 d old rice seedlings of Jiudao 51 (J-51) and Dongdao 4 (D-4) grown for 6 days under unstressed control (CK, nutrient solution with non-ABA), 100 mM NaCl (S0, S1, S5) and 15 mM $Na_2CO_3$ (A0, A1, A5) with the root-drenched priming of 0, 1 and 5 μM ABA for 24 h, respectively. Values are means ± SD, *n* = 3. Different letters on the column represent significant differences based on Ducan's test (*p* < 0.05).

However, under alkaline stress, PC1 and PC2 in J-51 accounted for 71.1% and 17.2% of the total variance, respectively, while PC1 and PC2 in D-4 accounted for 72.4% and 15.2%, respectively. PC1 reflected the variation between non-alkaline and alkaline conditions, while PC2 exhibited the difference between ABA-treated and non-ABA-treated rice seedlings. The growth-related physiological parameters (plant height and root DW), mineral traits (shoot and root $Na^+$ content and root $Na^+/K^+$ ratio), and enzymatic antioxidants (SOD, POD, CAT, and APX) were linked with ABA-treated J-51 rice seedlings under alkaline stress, whereas the mineral traits (shoot $Na^+/K^+$ ratio and shoot $Na^+$ content) and enzymatic antioxidants (CAT, APX, SOD, and POD) were linked with ABA-treated D-4 rice seedlings under alkaline stress (Figure 6C,D).

### 3.7. ABA Priming Mediated the Expression of Relevant Genes under Saline and Alkaline Stresses

The expression levels of genes related to ABA, stress tolerance, ion homeostasis, and reactive oxygen species scavenging were assessed in the leaves (Figure 7A,B). Compared with levels in CK, the transcript levels of *OsWsi18*, *OsSalT*, and *OsRbohA* were significantly elevated in J-51 and D-4 under both saline and alkaline stresses.

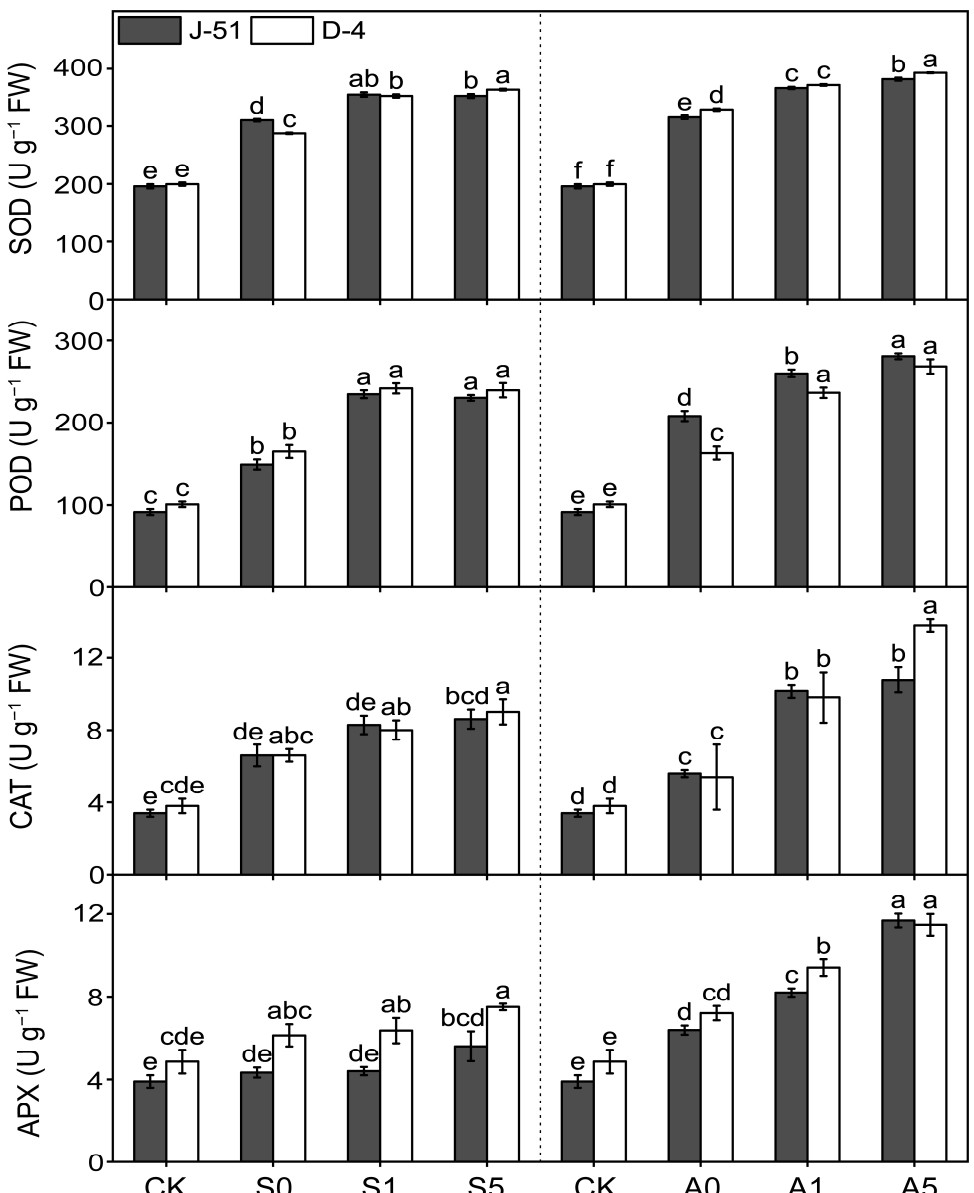

**Figure 5.** Effects of ABA priming on activities of active oxygen scavenging enzymes in rice seedlings under saline and alkaline stresses conditions. Enzyme activities of superoxide (SOD), peroxidase (POD), catalase (CAT) and ascorbate peroxidase (APX) in leaves of 14 d old rice seedlings of Jiudao 51 (J-51) and Dongdao 4 (D-4) grown for 6 days under unstressed control (CK, nutrient solution with non-ABA), 100 mM NaCl (S0, S1, S5) and 15 mM $Na_2CO_3$ (A0, A1, A5) with the root-drenched priming of 0, 1 and 5 μM ABA for 24 h, respectively. Values are means ± SD, *n* = 3. Different letters on the column represent significant differences based on Ducan's test (*p* < 0.05).

Under saline stress, 1 and 5 μM ABA pretreatment of J-51 significantly increased the expression of *OsHKT1;5*, *OsNAC9*, *OsSOS1*, *OsNHX5*, *OsPOX1*, *OsCATA*, *OsCu/Zn-SOD*, *OsNCED2*, *OsNCED3*, *OsWsi18*, and *OsSalT*, and decreased the expression of *OsRbohA* compared to levels in the non-ABA treatment. Pretreatment with 1 and 5 μM ABA in D-4 significantly upregulated *OsHKT1;5*, *OsSOS1*, *OsPOX1*, *OsAPX1*, *OsNCED3*, *OsWsi18*, and *OsSalT* and downregulated *OsRbohA* compared to levels under non-ABA treatment.

Under alkaline stress, 1 and 5 μM ABA pretreatment of J-51 significantly upregulated *OsHKT1;5*, *OsNAC9*, *OsSOS1*, *OsNHX5*, *OsPOX1*, *OsAPX1*, *OsNCED2*, *OsNCED3*, *OsWsi18*, and *OsSalT* and downregulated *OsRbohA* compared to levels under non-ABA treatment. Pretreatment with 1 and 5 μM ABA in D-4 significantly upregulated *OsHKT1;5*, *OsSOS1*,

*OsNHX5*, *OsAPX1*, *OsCATA*, *OsPOX1*, *OsNCED2*, *ONCED3*, *OsWsi18*, and *OsSalT* and downregulated *OsRbohA* compared to levels under non-ABA treatment.

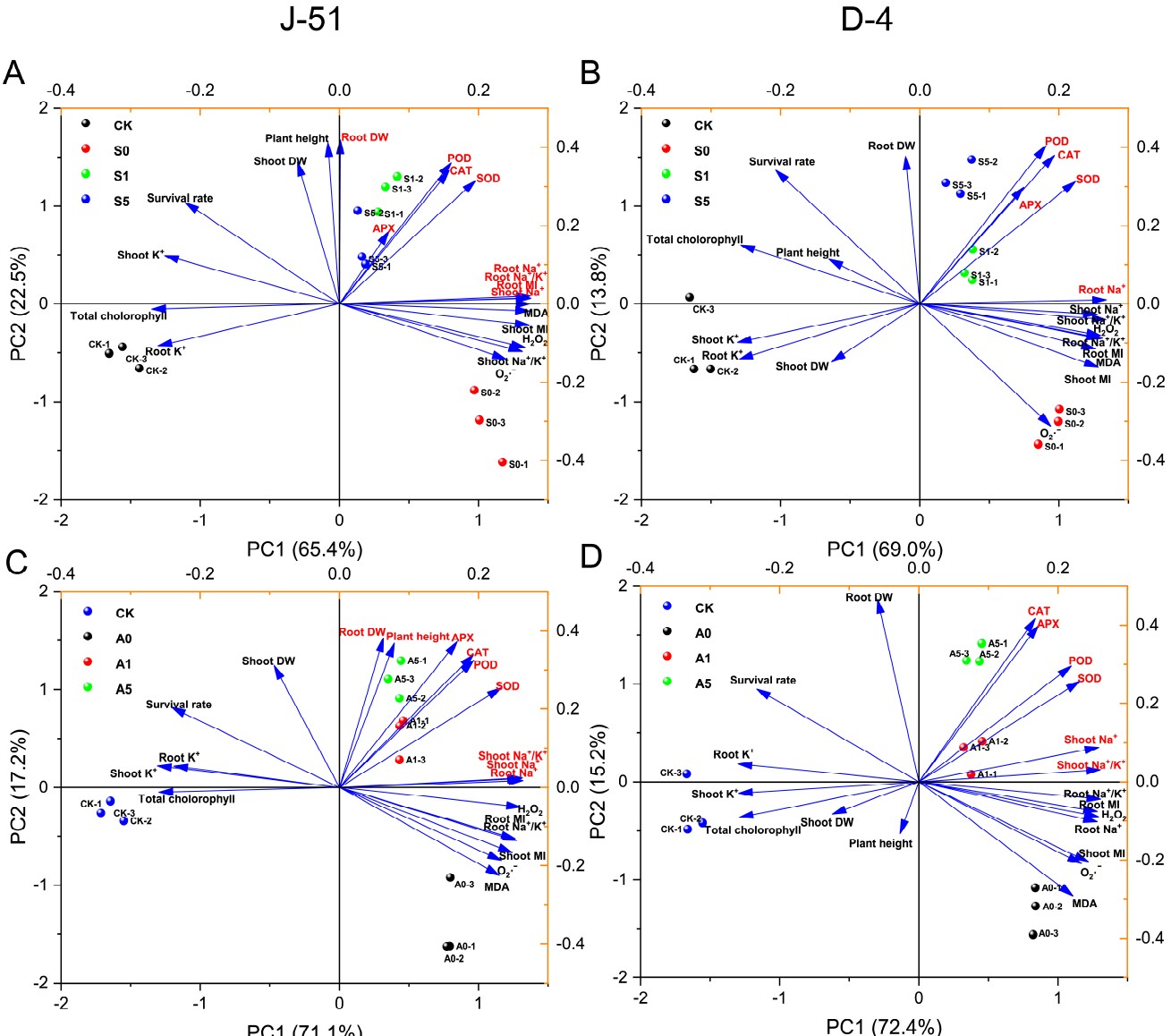

**Figure 6.** Principal component analysis (PCA) of different growth, physiological and biochemical parameters in rice seedlings under saline and alkaline stresses conditions with or without ABA priming. Principal component analysis (PCA) biplots of all growth and physiological parameters tested in Jiudao 51 (J-51) under saline stress (**A**) and alkaline stresses (**C**); Dongdao 4 (D-4) under saline stress (**B**) and alkaline stresses (**D**). The 14 d old rice seedlings of J-51 and D-4 grown under unstressed control (CK, nutrient solution with non-ABA), 100 mM NaCl (S0, S1, S5) and 15 mM $Na_2CO_3$ (A0, A1, A5) with the root-drenched priming of 0, 1 and 5 μM ABA for 24 h, respectively.

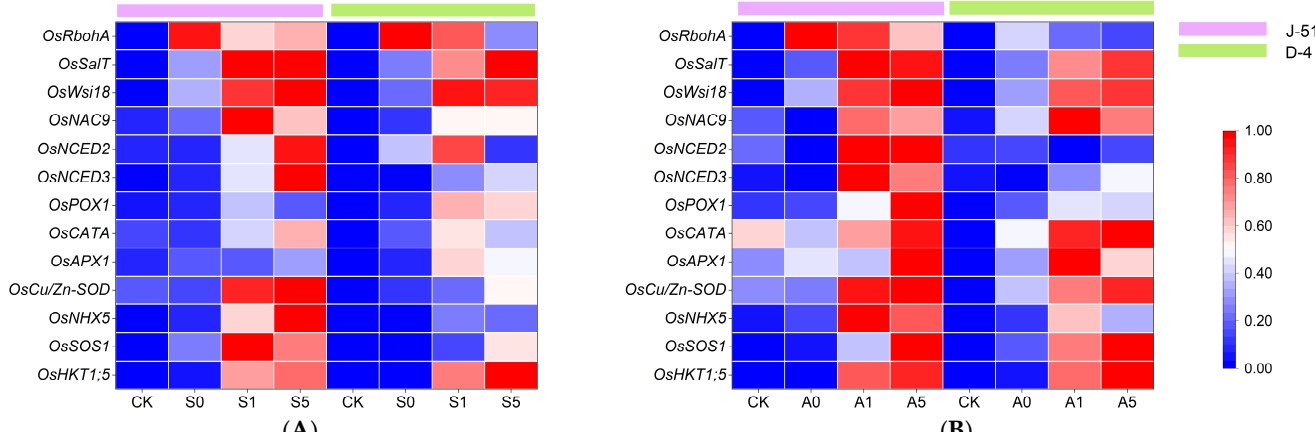

**Figure 7.** Expression levels of genes related to ABA, stress tolerance, ion homeostasis, reactive oxygen species production and scavenging. ABA response-related genes *OsSalT* and *OsWsi18*, stress tolerance-related gene *OsNAC9*, ions homeostasis-related genes *OsHKT1;5*, *OsSOS1*, and *OsNHX5*, reactive oxygen species scavenged-related genes *OsPOX1*, *OsCATA*, *OsAPX1*, and *OsCu/Zn-SOD*, reactive oxygen species production gene *OsRbohA*, and ABA biosynthesis genes *OsNCED2* and *OsNCED3* in leaves of 14 d old rice seedlings of Jiudao 51 (J-51) and Dongdao 4 (D-4) under saline stress (**A**) and alkaline stresses (**B**). The 14 d old rice seedlings of J-51 and D-4 grown under unstressed control conditions (CK, nutrient solution with non-ABA), 100 mM NaCl (S0, S1, S5), and 15 mM $Na_2CO_3$ (A0, A1, A5) with the root-drenched priming of 0, 1, and 5 μM ABA for 24 h.

## 4. Discussion

Saline–alkaline soil has become a significant environmental stressor that hinders plant growth and productivity via the complex interplay between salinity, alkalinity (high pH), and osmotic pressure [6]. Saline and alkaline stresses strongly influence seed germination and plant growth in rice, thereby reducing plant height and inhibiting metabolic processes [54]. The development of effective and efficient strategies to enhance rice plant tolerance to SA stress is crucial for improving plant growth and productivity in SA fields.

The tolerance of plants to future stressors can be enhanced through exposure to stress, the application of plant hormones and their functional analogs [55], the utilization of plant metabolites [56], mineral salts, and reactive chemicals [57]. This process is known as priming [35,36,58]. The priming of plants results in more efficient plant responses to subsequent stress, and this primed state can persist for extended periods of time, even across generations [58]. Examples of well-known priming mechanisms include systemic acquired resistance and seed pre-conditioning (seed priming) [59]. ABA mediates plant resistance to environmental stress [60–62]. This hormone exerts a priming effect on plants by activating defense mechanisms and signaling pathways [40,41]. Despite many studies focused on ABA resistance to various abiotic stresses, relatively little is known about the priming effect of ABA on plant tolerance to saline and alkaline stresses.

We demonstrated rice seedling growth under different conditions and found that after emergence, rice seedlings grow normally under the CK condition, while saline and alkaline stresses cause leaf curling and withering, inhibit the formation of new leaves and roots, and even lead to the death of J-51 and D-4 plants as the stress duration increases (Figure 1A,B). Furthermore, alkaline stress exerted greater effects on survival and growth than those saline stress, particularly in J-51 (a cultivar sensitive to SA stress) (Figures 1 and 2). These observations are consistent with those of Lv et al. [6] and Liu et al. [63]. In this study, we demonstrated that pretreatment with exogenous ABA significantly augmented the survival rate and chlorophyll content of rice seedlings under both saline and alkaline stress conditions (Figures 1 and 2). Similar to our results, Chen et al. [62] reported that ABA enhanced the photosynthetic capacity of rice by protecting photosynthetic pigments, as evidenced by delayed leaf senescence to mitigate salt stress-induced damage. The alteration

of the photosynthetic pigment composition represents a crucial adaptive mechanism in plants under adverse environmental conditions [64]. The decrease in plant productivity under salt stress is often attributed to reduced photosynthetic rates resulting from a decline in the concentration of photosynthetic pigments [65].

In addition, there are some indications that ABA biosynthesis could be slightly enhanced by the overexpression of *OsNCED2* and *OsNCED3* under stress conditions (Figure 6); these genes encode key enzymes in the ABA biosynthetic pathway of rice that regulate various processes, such as seed dormancy, plant growth, stress tolerance, and leaf senescence [66]. *OsSalT* and *OsWsi18*, two ABA-responsive genes [67], were significantly induced by ABA pretreatment in this study. The upregulation of these ABA-response genes may enhance ABA signaling, resulting in an effect similar to that of exogenous ABA pretreatment under saline and alkaline stresses. Moreover, the significant upregulation of the stress-responsive gene *OsNAC9* may explain a higher tolerance of seedlings to alkaline stress than to saline stress. The data presented herein demonstrate that ABA exerts a robust priming effect on the adaptive response to saline and alkaline stresses in rice seedlings, with a more pronounced induction of resistance to alkaline stress conditions. Furthermore, this priming effect was particularly beneficial for SA-sensitive cultivars.

Plants growing in saline soils are susceptible to ion toxicity [68]. Almeida et al. identified that a significant elevation in the $Na^+$ concentration was the primary cause of salt-induced damage [69]. To maintain the shoot $Na^+/K^+$ balance, plants take up more $K^+$ and exclude toxic $Na^+$. In the present study, ABA pretreatment had the opposite effect; rice seedlings treated with ABA under saline and alkaline stresses showed a reduced $Na^+$ content but increased $K^+$ content, resulting in a lower $Na^+/K^+$ ratio than that of untreated rice seedlings under stress conditions. Marusig and Tombesi [40] demonstrated that the exogenous application of ABA can regulate leaf stomatal conductance by inhibiting $Na^+$ accumulation in leaves, consistent with the findings of this study. These findings were consistent with those obtained from the PCA, which revealed that the $Na^+$ content and the $Na^+/K^+$ ratio in the roots were positively correlated with ABA-treated rice seedlings under saline stress conditions, whereas the $Na^+$ content and $Na^+/K^+$ ratio in the shoots were positively associated with ABA-treated rice seedlings exposed to alkaline stress. In addition, the increases in the $Na^+$ content and $Na^+/K^+$ ratio in the roots were more pronounced under alkaline stress than under saline stress, whereas saline stress resulted in a greater elevation in the $Na^+$ content and $Na^+/K^+$ ratio in the shoots than that induced by alkaline stress. This discrepancy may be attributed to the differential sensitivity of rice seedling shoots and roots towards saline and alkaline stresses.

Rice high-affinity potassium ($K^+$) transporter 1;5 (OsHKT1;5) maintains $Na^+/K^+$ homeostasis under salt stress as a $Na^+$-selective transporter that mediates $Na^+$ exclusion from the shoots via $Na^+$ removal from the xylem sap [1]. Wang et al. [70] isolated a salt stress-sensitive mutant *osbag4-1*, showing a significant reduction in the expression of *OsHKT1;5* and the $K^+$ content and a significant increase in the $Na^+$ level in shoots. Salt overly sensitive protein 1 (SOS1) and $Na^+/H^+$ exchanger (NHX) also play key roles in $Na^+$ homeostasis (including $Na^+$ absorption and distribution) [71]. Exposure to high-salinity stress triggers the initiation of a calcium signal that subsequently activates the SOS pathway [72]. The expression of *SsSOS1* is significantly induced by NaCl in *Suaeda salsa* [13]. *OsHKT1;5*, *OsSOS1*, and *OsNHX5* were significantly upregulated after ABA pretreatment at 1 and 5 μM compared with levels under stress conditions alone in this study. Therefore, the findings from this study, along with those previous reports, provide further support for the crucial role of ABA in enhancing saline and alkaline tolerance in rice via the maintenance of ion homeostasis.

Plants produce ROS, including $O_2 \cdot^-$ and $H_2O_2$, in response to various abiotic stresses, resulting in cellular toxicity and eliciting detrimental effects on multiple cellular compartments, including lipid peroxidation, nucleic acid modification, and the inhibition of crucial enzyme activities [73]. In this study, the exposure of rice seedlings to saline and alkaline stresses resulted in significant elevations in $O_2 \cdot^-$ and $H_2O_2$ levels that coincided with an in-

crease in the MDA content and MI percentage of rice seedlings (Figure 4A,B). The degree of elevation was significantly more pronounced under alkaline stress than under saline stress, which may have induced greater detrimental effects on plant growth. On the contrary, pretreatment with 1 and 5 μM ABA effectively mitigated ROS accumulation and reduced the MI percentage in rice seedlings subjected to saline and alkaline stresses (Figure 4A,B), highlighting the crucial role of ABA in priming rice seedlings against oxidative damage induced by salinity and alkalinity. The results of our study are consistent with those of previous research demonstrating the inhibitory effects of ABA on the accumulation of $H_2O_2$ and $O_2 \cdot^-$ in rice leaves under high-alkaline conditions [74].

The decrease in ROS accumulation coincided with increases in SOD, CAT, APX, and POD activities in the leaves of ABA-treated rice seedlings under saline and alkaline stresses (Figure 5). Similarly, a PCA revealed a positive correlation between ABA treatment and SOD, CAT, APX, and POD activities under saline and alkaline stresses (Figure 6A–D). These findings suggest that ABA effectively functions as a protective mechanism by enhancing the antioxidant defenses against oxidative damage in rice plants exposed to saline and alkaline stresses. This is corroborated by previous research demonstrating that ABA application can augment the activities of antioxidant enzymes in alfalfa subjected to alkaline stress [52] and in plants exposed to heat stress [75]. The influence of salinity and alkalinity on cells and cellular compartments can be mitigated through modifications in cytoskeleton functioning, membrane properties (including proteins), and other adaptations at the cellular and tissue levels [76]. Overall, ABA pretreatment significantly augmented the activity of antioxidant enzymes, serving as a crucial defense mechanism against stress-induced lipid peroxidation and membrane impairment in diverse plant species. Notably, the increases in levels of *OsCu/Zn-SOD*, *OsAPX1*, *OsCATA*, and *OsPOX1* were higher in rice seedlings exposed to alkaline stress than in those subjected to saline stress with ABA pretreatment (Figure 7).

Saline and alkaline stresses disrupt numerous biochemical and physiological processes, resulting in a significant reduction in the growth performance of rice seedlings. Root-drenched ABA priming significantly alleviated the damage caused by saline and alkaline stresses. However, the extent and manner of mitigation may differ. Rice seedlings pretreated with ABA under alkaline stress conditions showed a greater improvement in the chlorophyll content and activities of ROS scavenging enzymes (SOD, POD, CAT, and APX) and a lesser improvement in the shoot $K^+$ content than those under saline stress conditions. Additionally, they showed significantly higher reductions in root $Na^+$, MDA, $H_2O_2$, and $O_2 \cdot^-$ contents, cell membrane integrity (MI), and the root $Na^+/K^+$ ratio and lower reductions in the shoot $Na^+$ content and $Na^+/K^+$ ratio than those under saline stress conditions. Furthermore, ABA priming under alkaline stress had a stronger ability to maintain ion homeostasis by superinducing the expression of *OsHKT1;5*, *OsSOS1*, and *OsNHX5*, resulting in a lower $Na^+$ content and $Na^+/K^+$ ratio than those in rice seedlings with ABA pretreatment under saline stress. Rice seedlings with ABA priming under alkaline stress had a stronger ability to eliminate ROS by significantly upregulating *OsPOX1* and *OsCATA*, resulting in lower increases in the MDA, $H_2O_2$, and $O_2 \cdot^-$ contents, and lower decreases in ROS-related enzyme activities in both cultivars compared with those in rice seedlings with ABA pretreatment under saline stress. The increased levels of *OsNCED3*, *OsSalT*, and *OsWsi18* in ABA-treated rice seedlings were significantly higher under alkaline stress than under saline stress, suggesting that stress-induced changes in endogenous ABA levels were triggered by exogenous ABA pretreatment under alkaline stress. Downregulation of the ROS-producing gene *OsRbohA* under alkaline stress was significantly greater than that under saline stress, suggesting that ABA pretreatment can more effectively mitigate oxidative damage caused by alkaline stress (Figure 8).

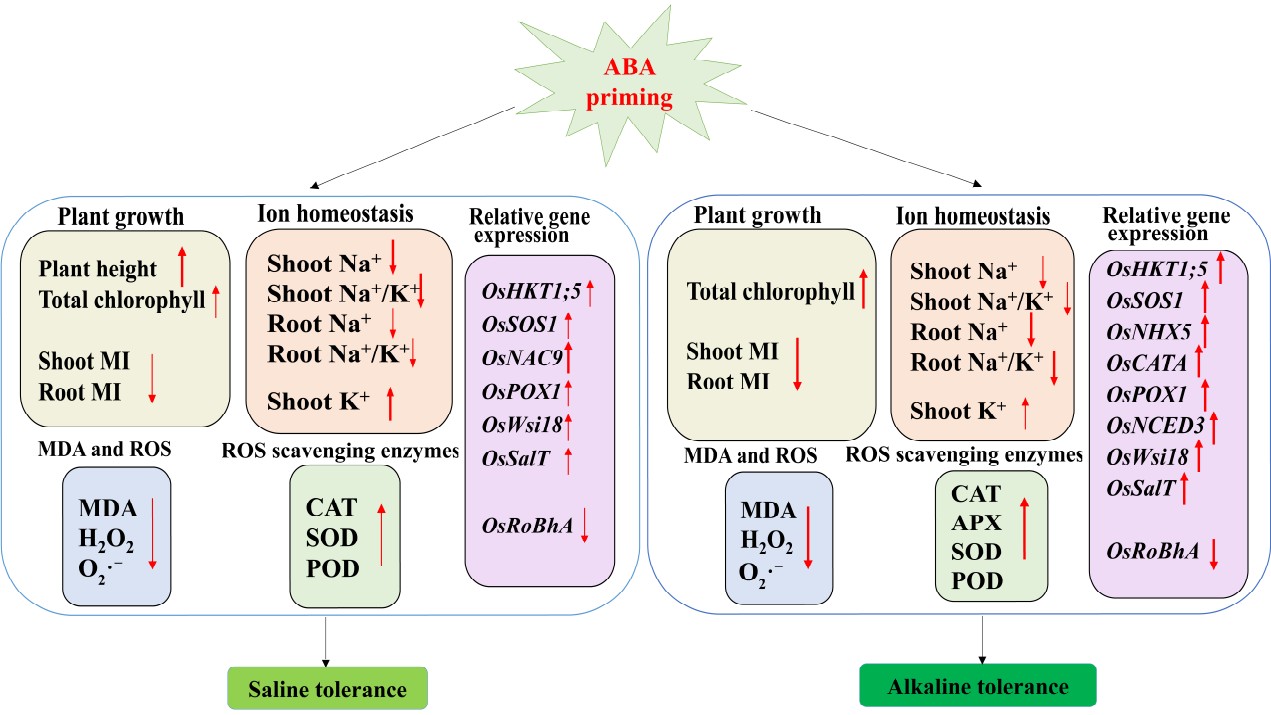

**Figure 8.** Proposed model illustrating the mechanisms by which ABA priming enhances saline and alkaline stress tolerance in rice. The application of ABA resulted in significantly greater improvements in the chlorophyll content and activities of ROS scavenging enzymes (SOD, POD, CAT and APX) and lower improvements in the shoot $K^+$ content in rice seedlings under alkaline stress conditions than under saline stress. Additionally, there were significantly greater reductions in the content of MDA, levels of ROS ($H_2O_2$, and $O_2\cdot^-$), cell membrane integrity (MI), content of $Na^+$ and $Na^+/K^+$ ratio of roots, and lower reductions in content of $Na^+$ and $Na^+/K^+$ ratio of shoots compared with those under saline stress. Furthermore, rice seedlings with ABA pretreatment under alkaline stress had a stronger ability to maintain ion homeostasis, eliminate ROS, and induce changes in endogenous ABA levels via the upregulation of *OsHKT1;5*, *OsSOS1*, *OsNHX5*, *OsPOX1*, *OsCATA*, *OsNCED3*, *OsSalT*, and *OsWsi18*, and downregulation of *OsRbohA*, resulting in reduced increases in content of $Na^+$ and $Na^+/K^+$ ratio of shoots, content of MDA, levels of ROS ($H_2O_2$, and $O_2\cdot^-$), and MI of shoots and roots and smaller reductions in the total chlorophyll content and activities of ROS scavenging enzymes (SOD, POD, CAT, and APX) in both cultivars compared with those in rice seedlings with ABA pretreatment under saline stress. Note: Upward and downward red arrows for the two stress types indicate an increase and decrease in parameters with ABA pretreatment compared to corresponding values under saline stress and alkaline stress conditions (100 mM NaCl and 15 mM $Na_2CO_3$), respectively. The thickness of the red arrows represents the relative changes in parameters after ABA priming between the two types of stresses.

## 5. Conclusions

Taken together, ABA priming has been recognized as an environmentally friendly agrobiological approach that enhances tolerance to saline and alkaline stresses through its involvement in diverse biochemical and physiological processes, while regulating the expression of genes related to ABA signaling, stress tolerance, ion homeostasis, and ROS production and scavenging. Furthermore, rice seedlings with ABA pretreatment under alkaline stress had a stronger ability to maintain ion homeostasis, eliminate ROS, and induce changes in endogenous ABA levels via the upregulation of *OsHKT1;5*, *OsSOS1*, *OsNHX5*, *OsPOX1*, *OsCATA*, *OsNCED3*, *OsSalT*, and *OsWsi18*, and downregulation of *OsRbohA*, thereby attenuating the increases in the $Na^+$ content, $Na^+/K^+$ ratio, contents of MDA, $H_2O_2$, $O_2\cdot^-$, and MI of shoots and roots, and preventing the reductions in the total chlorophyll content and ROS scavenging enzymes activities (SOD, POD, CAY, and APX)



in both cultivars compared with estimates in rice seedlings with ABA pretreatment under saline stress. The SA-sensitive cultivar demonstrated greater sensitivity to the priming effect of ABA than that of the SA-tolerant cultivar under both stress conditions. The findings of this study have significant implications for the adaptation of rice to SA soils.

**Supplementary Materials:** The following supporting information can be downloaded at https://www.mdpi.com/article/10.3390/agronomy13112698/s1. Table S1: List of primers used in quantitative real-time PCR analysis.

**Author Contributions:** W.L. and Z.L. designed and planned the research. Z.F., G.L., M.S., Y.J., Y.X., X.L., Y.G., T.Y. and J.H. performed experiments and analyzed the data. M.W., M.L., H.Y. and Z.X. helped with designing, executing and interpreting experiments. Z.F., G.L., Z.L. and W.L. edited the manuscript. W.L. and Z.L. agree to serve as the author responsible for contact and communication. All authors have read and agreed to the published version of the manuscript.

**Funding:** This research was funded by the Strategic Priority Research Program of the Chinese Academy of Sciences (Project No. XDA28110105, XDA28110102); the National Key Research and Development Program of China (Project No. 2022YFD1500505); Science and Technology Project of Education Department of Jilin Province (Project No. JJKH20230019KJ); Science and Technology Development Plan Project of Baicheng city (Project No. 202206); the Technology Cooperation High-Tech Industrialization Project of Jilin Province and the Chinese Academy of Sciences (Project No. 2023SYHZ0045).

**Informed Consent Statement:** The authors declare no conflict of interest.

**Data Availability Statement:** The data presented in this study are available on request from the corresponding author.

**Conflicts of Interest:** The authors declare that the research was conducted in the absence of any commercial or financial relationships that could be construed as a potential conflict of interest.

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
