# Peer review of "Comparative Study of the Priming Effect of Abscisic Acid on Tolerance to Saline and Alkaline Stresses in Rice Seedlings"

_agronomy, doi:10.3390/agronomy13112698_

Round 1
Reviewer 1 Report
Article Comparative Study of the Priming Effect of Abscisic Acid on Tolerance to Saline and Alkaline Stresses in Rice Seedlings by Zhonghui Feng, Guanru Lu, Miao Sun, Yangyang Jin, Yang Xu, Xiaolong Liu, Mingming Wang, Miao Liu, Haoyu Yang, Yi Guan, Tianhe Yu, Jiafeng Hu, Zhiming Xie, Weiqiang Li, Zhengwei Liangof considers the issues of physiological, biochemical, and molecular investigations to analyze the characteristics and reactions of two cultivars of rice under saline and alkaline stresses.
The manuscript is formatted according to the rules of the journal and contains all the necessary sections.
However, there are a number of questions.
Firstly, the introduction does not discuss a number of issues regarding the physiological responses of plants to the stress influences under study. It is especially alarming that the authors do not use the term sodisity. Perhaps a methodological problem is also associated with this, namely that to simulate toxic salinity of alkalinity, the authors used a rather toxic but not typical for natural conditions compound Na2CO3, while in soda salinity NaHCO3 is more often found. This is due to the fact that the first compound often goes into an insoluble carbonate form, while the second is present in the form of ions and can have severe toxic effects. The authors should explain the reason for choosing a rather controversial methodology for this part of the study, but an example for the purpose of coarsening the results.
It is equally important to discuss the results in the context of the most significant works related to the analysis of the causes of damage to cells and cellular compartments of plants, for example: Munns R., Gilliham M. Salinity tolerance of crops—what is the cost? //New phytologist. – 2015. – T. 208. – No. 3. – P. 668-673; Flowers T. J., Colmer T. D. Salinity tolerance in halophytes //New phytologist. – 2008. – P. 945-963; Baranova E. N., Gulevich A. A. Asymmetry of plant cell divisions under salt stress //Symmetry. – 2021. – T. 13. – No. 10. – S. 1811; Safdar, H., Amin, A., Shafiq, Y., Ali, A., Yasin, R., Shoukat, A., ... & Sarwar, M. I. (2019). A review: Impact of salinity on plant growth. Nat. Sci, 17(1), 34-40.
Restriction of growth due to sodicity and salinity, as well as changes in acidity, are associated with disruption of the functioning of plastids and mitochondria and with proliferation and elongation growth. This is overcome by changes in the functioning and modification of the cytoskeleton, membranes and membrane proteins, and other changes at the level of cells and tissues.
For this reason, I recommend expanding the literary analysis of the premises in both the introduction and discussion. To do this, it is desirable to expand the description of the details of impacts and effects in the articles mentioned by the authors. The authors probably shortened the original version, since on line 43 even the list of cited sources is indicated with an error and in the wrong sequence.
The description of the soil conditions that the authors write about should be considered specifically, indicating the ions and reasons for the occurrence of this type of salinization - primary and/or secondary, respectively.
Some notes apply to the drawings
Photo 1a should be enlarged and a ruler added.
Figure 1b and all histograms up to and including Figure 5 should be replaced or improved. The reason is that the confidence intervals are not visible on the black image. Make the histograms gray or colored.
Figure 6 cannot be called successful, since the differences are not highlighted in any way; you need to look at the small inscriptions and look for analogues, which is difficult with a small image. It might be good to use color or a sign to indicate clear differences with a high level of significance.
The reason for the separation of ROS and ROS metabolism in Figure 8 is obvious to me, I think we need to rename metabolism to enzymes.
In addition, the logical construction of the drawing is not correct. It turns out that ABA causes salinity and alkalinity stress, which is not true.
Minor notes: the authors should check the spelling of the indices for ions, since they are executed differently, which is not correct, for example: pp. 463, 458
I would also like to see data on germination and effects on early morphology.
The rest of the work is done carefully and neatly and can be published after correction of comments.
Reviewer 2 Report
The manuscript entitled "Comparative Study of the Priming Effect of Abscisic Acid on Tolerance to Saline and Alkaline Stresses in Rice Seedlings" is a nice attempt to study the response of ABA in seed germination under saline and alkali conditions.
ABA biosynthesis is a multistep pathway in higher plants and accumulation of ABA regulates various seed activities. Usually seed priming is performed to enhance seed vigour in terms of germination potential and increased stress tolerance without knowing proper mechanisms and its actions.
The present manuscript infers various genes as well as their role in seed metabolism, reduces oxidative stress and maintains ion homeostasis.
There are few suggestions for improvement of the manuscript.
Page No. 2 Line No. 70, The objective one says that the experiments will infer different responses of cultivars towards salinity and alkali stress?? whereas within two genotypes they showed contrasting results.
There is a need to study yield and quality related traits due to this positive effect of seed priming with ABA.
Reviewer 3 Report
I have only one suggestion about Figures. Figures 2-5 must be uniform (same font, colours...). Figure 6: not clear, change font number.
Minor editing of English language required. Some sentences must be clearly.
Round 2
Reviewer 1 Report
Manuscript Comparative Study of the Priming Effect of Abscisic Acid on Tolerance to Saline and Alkaline Stresses in Rice Seedlings
by Zhonghui Feng , Guanru Lu , Miao Sun , Yangyang Jin , Yang Xu , Xiaolong Liu, Mingming Wang, Miao Liu, Haoyu Yang, Yi Guan, Tianhe Yu, Jiafeng Hu, Zhiming Xie, Weiqiang Li, Zhengwei Liang addresses issues of rice tolerance.
Text and images have been corrected.
The authors have made their adjustments and the manuscript is ready for publication in its present form.